# Determination by ICP-MS of Essential and Toxic Trace Elements in Gums and Carrageenans Used as Food Additives Commercially Available in the Portuguese Market

**DOI:** 10.3390/foods12071408

**Published:** 2023-03-26

**Authors:** Rui Azevedo, Ana Rafaela Oliveira, Agostinho Almeida, Lígia Rebelo Gomes

**Affiliations:** 1LAQV/REQUIMTE, Department of Chemical Sciences, Faculty of Pharmacy, University of Porto, 4050-313 Porto, Portugal; 2FP-I3ID, University Fernando Pessoa, 4249-004 Porto, Portugal; 3LAQV/REQUIMTE, Department of Chemistry and Biochemistry, Faculty of Sciences, University of Porto, 4169-007 Porto, Portugal

**Keywords:** food additives, gums, carrageenans, ICP-MS, trace elements, heavy metals

## Abstract

Gums and carrageenans are food additives widely used in food preparations to improve texture and as viscosifiers. Although they are typically added in small amounts, nowadays people tend to use more and more pre-prepared food. In this work, the content of a wide panel of trace elements in commercial products were analyzed. Carrageenans and gums (n = 13) were purchased in the Portuguese market and were from European suppliers. Samples were solubilized by closed-vessel microwave-assisted acid digestion and analyzed by ICP-MS. Globally, the content of essential trace elements decreased in the following order: Fe (on average, on the order of several tens of µg/g) > Mn > Zn > Cr > Cu > Co > Se > Mo (typically < 0.1 µg/g), while the content of non-essential/toxic trace elements decreased in the following order: Al > Sr > Rb > As > Li > Cd > Pb > Hg. The consumption of these food additives can significantly contribute to the daily requirements of some essential trace elements, namely Cr and Mo. The toxic trace elements Cd, As, Pb, and Hg were below the EU regulatory limits in all analyzed samples. Additional research is needed to define the potential risk of introducing toxic trace elements into food products through the use of these additives.

## 1. Introduction

In order to meet the nutritional needs of the growing world population, increasing food production and reducing food waste seems to be imperative. This objective is challenging because it is necessary to find sustainable production processes that respond to the increase in demand and, at the same time, comply with the current safety and quality requirements. Food additives allow manufacturers to extend food’s shelf life and thus contribute to reducing food waste and supporting the nutritional sustainability of the global population. Gums and carrageenans are food additives widely used in food preparations to improve texture and act as viscosifiers.

Refined (E407) or semi-refined (E407a) carrageenans are important commercial hydrophilic colloids (water-soluble gums) that occur as matrix material in numerous species of red seaweeds (Rhodophyta). In the production of semi-refined carrageenans (SRC), the seaweed is washed in an alkaline solution in order to cause desulfation of the galactose units of the carrageenan, leading to the desired galactose polymers by dehydration and reorientation. After neutralization and washing in water, the seaweed is dried, leaving behind carrageenans and other insoluble matter. Thus, SRC consist largely of carrageenan and cellulose. Additional treatments such as filtration and purification can be applied to remove some residual substances such as cellulosic materials, thus forming refined carrageenans (RC). Carrageenans can be classified as kappa (κ), lambda (λ) or iota (ι), depending on molecular weight. The product extracted from seaweed is generally composed of a mixture of κ- and λ-carrageenans, as they are in different stages of the reproductive cycle. Some seaweed species, such as *Gymnogongrus furcellatus*, have been used as a source of ι-type carrageenans, but the stocks of this algae are smaller. Carrageenans are used to gel, thicken, or suspend. They are used to stabilize emulsions, to control syneresis, and as bulking, binding, and dispersion agents. Their main use in food is in dairy-free preparations. In Europe, κ-carrageenans have been used as a powder for topping pies and cakes and as an ingredient in milk pudding. Carrageenans are unique in their ability to suspend, at very low concentration (ca. 300 ppm), cocoa in chocolate milk; no other gum has shown such great ability. The use of ι-carrageenans in gel desserts provides products with textures very similar to gelatin gels. They even have an advantage over gelatin gels due to their higher melting point, so they find a particular market in tropical climates or where refrigeration systems are not widely available. Another advantage is that ι-carrageenan gels maintain their soft structure with age, whereas gelatin tends to harden. This is especially important for ready-to-eat desserts.

Generally, gums are used to stabilize formulations; they improve the melting capacity of ice-based preparations and are used to thicken soups, baked goods, yoghurt, milk, and cheese. Guar gum (E412) is the ground endosperm of the seeds of *Cyamopsis tetragonoloba* L. Taub., and has been used to create processed foods, acting as a thickener and preventing the formation of ice crystals. Guar is also an indispensable ingredient in baked goods due to its unique ability to improve crumb texture and structure and contribute to shelf-life extension. Acacia (Arabic) gum (E414) is a dried exudate obtained from the stems and branches of natural strains of *Acacia senegal* L. Willdenow or closely related Acacia species. It exhibits emulsifying, stabilizing, binding, and shelf-life-enhancing properties. Xanthan gum (E415) is a high molecular weight polysaccharide produced by fermenting a carbohydrate with a pure culture of *Xanthomonas campestris* strains. It is used to prevent oil separation by stabilizing emulsions (although it is not an emulsifier), and also helps to suspend solid particles such as spices. Tara gum (E417) is a high viscosity polysaccharide isolated from the endosperm of *Caesalpinia spinosa* L. seeds. Tara gum adds texture to creams, increases the free/thawed stability in frozen desserts, liquid dairy products, creams, and puddings, and adds viscosity to beverages. It also reduces syneresis in jams, jellies, and fillings, and improves mouthfeel. Gellan gum (E418) is a high molecular weight polysaccharide gum obtained from the fermentation of a carbohydrate by strains of *Pseudomonas elodea* (now *Sphingomonas elodea*). It is used to create gels, add texture, stabilize and suspend ingredients or nutrients, form films, and create structure. These properties make gellan gum a good choice for products with innovative textures, where nutrients or ingredients must remain blended, or that are served in unique shapes in molecular gastronomy.

Although additives are typically used in small amounts in foods (for example, 2% of a gum can be enough to gel a food preparation), nowadays people tend to use more and more pre-prepared foods. In addition, there is no legal obligation to inform the consumer about the amount of additive used in a given formulation. This highlights the importance of considering the cumulative effects associated with the daily intake of different types of additives, a concern of the European Food Safety Authority (EFSA) that is reflected in the re-evaluations of food additives that have been carried out in recent years [1,2,3,4,5,6].

The World Health Organization (WHO) considers that at least 19 elements can significantly affect human health and classifies them into three categories [7]: (i) Proven essential elements: including Cr, Fe, Co, Cu, Zn, Se, and Mo. They must be ingested in small amounts throughout life and the lack of sufficient intake triggers various diseases; (ii) Elements that may be beneficial, which include Mn and Sn; (iii) Possibly toxic elements: including Li, Al, As, Cd, Pb and Hg.

To our knowledge, there are no published studies on the trace element content of carrageenan and gums used as food additives. The EFSA regularly reassesses the safety of these food additives and has called for the revision of the maximum limits for toxic elements As, Cd, Hg and Pb [1,2,3,4,5,6]. According to the latest EFSA report on carrageenan-based food additives, As, Cd, Hg and Pb concentrations of the analyzed samples varied between 0.005–2.000 mg/kg, 0.005–1.090 mg/kg, 0.0001–1.0000 mg/kg, and 0.0001–2.12 mg/kg, respectively [6]. EFSA reports typically indicate lower concentrations of toxic trace elements in gums. In Acacia gum, all samples had concentrations of As, Cd, and Hg lower than 0.05 mg/kg, 0.01 mg/kg, and 0.008 mg/kg, respectively, while the Pb content ranged from <0.005 mg/kg to 0.048 mg/kg [2]. In guar gum, the concentration of toxic trace elements (provided by the industry) varied between <0.005 and <0.01 mg/kg for As, <0.01 and <0.1 mg/kg for Cd, <0.01 and <0.1 mg/kg for Hg, and <0.02 and 0.21 mg/kg for Pb [1].

In this work, the contents of a wide group of trace elements in commercial samples of gum and carrageenans were measured by inductively coupled plasma spectrometry (ICP-MS). It was intended to provide data that could contribute to completing the risk assessment studies that have been carried out, namely with regard to the levels of the main toxic trace elements (i.e., Pb, Hg, Cd and As), as well as others considered essential for humans but which can become toxic depending on the level of daily intake. The main objective was to evaluate their nutritional value and especially their safety as a food additive.

## 2. Materials and Methods

### 2.1. Samples

Samples of carrageenans and gums were obtained from the Portuguese market between October 2021 and March 2022, and were from European suppliers. Eight carrageenans samples were analyzed: sample 1 was an SRC consisting of a mixture of κ and λ; samples 2–3 were an SRCκ; samples 4–5 were an SRCι; samples 6–7 were an RCκ; and sample 8 was an RCι. The other five samples were gums: 9 [Guar], 10 [Acacia (Arabic)], 11 [Xanthan], 12 [Tara], and 13 [Gellan]. The gums were all from the same supplier while the carrageenans were from three different suppliers.

### 2.2. Samples Solubilisation

Samples were solubilized by closed-vessel microwave-assisted acid digestion in an ETHOS™ EASY microwave oven (Milestone, Sorisole, Italy) equipped with an SK-15 EasyTEMP high-pressure rotor following a procedure based on a U.S. Environmental Protection Agency (EPA) method [8]. A sample mass of approximately 0.4 g was weighed directly into the PTFE-TFM (modified polytetrafluorethylene) vessels of the microwave oven and 9 mL of high-purity nitric acid (HNO_3_ ≥ 69%, TraceSELECT™, Honeywell Fluka, Seelze, Germany), 0.5 mL of high-purity hydrochloric acid (HCl ≥ 30%, TraceSELECT™, Honeywell Fluka), 0.5 mL of high-purity hydrofluoric acid (HF 47–51%, TraceSELECT™, Fluka, Steinhein, Germany), and 1 mL of high-purity hydrogen peroxide (H_2_O_2_ 30%, Suprapur^®^, Supelco, Darmstadt, Germany) were added. Digestion was performed using the following microwave oven program: gradual temperature increase for 20 min to 210 °C, followed by 15 min at 210 °C. After cooling to room temperature, the vessels were opened, the sample solutions were transferred to decontaminated 50 mL polypropylene tubes, and the volume was adjusted with ultrapure water (resistivity >18.2 MΩ.cm at 25 °C), obtained with an Arium^®^ pro water purification system (Sartorius, Göttingen, Germany), to a final volume of 20 mL. Sample blanks were obtained using the same procedure. The obtained solutions were stored at 4 °C until analysis. Each sample was run in triplicate, and a digestion blank was run on each digestion batch in a random vessel.

### 2.3. Trace Element Determination

The analysis of the solutions was performed by ICP-MS. The instrument was an iCAP™ Q (Thermo Fisher Scientific, Waltham, MA, USA), equipped with a Meinhard^®^ (Golden, CO, USA) TQ+ concentric quartz nebulizer, a Peltier-cooled high-purity quartz baffled cyclonic spray chamber, and a demountable quartz torch with a 2.5 mm i.d. quartz injector. The interface consisted of two (sampler and skimmer) Ni cones. High-purity argon (99.9997%), supplied by Gasin (Matosinhos, Portugal), was used as a nebulizer and plasma gas. Prior to each analytical run, the instrument was tuned for maximum sensitivity and signal stability, and for minimal formation of oxides and double-charge ions. The main operational parameters of the ICP-MS were: nebulizer gas flow, 1.14 L/min; auxiliary gas flow, 0.79 L/min; plasma gas flow, 13.9 L/min; radiofrequency generator power, 1550 W; and dwell time, 10 ms.

Sample solutions were diluted with a solution containing 2% *v*/*v* HNO_3_, 0.5% *v*/*v* HCl, 500 µg/L Au (Gold Standard for ICP, 1000 mg/L, TraceCERT^®^, Fluka), and 10 µg/L internal standard (IS; Periodic table mix 3, 10 mg/L, TraceCERT^®^, Sigma-Aldrich, St. Louis, MO, USA). For Li, Al, Cr, Mn, Co, Cu, Zn, As, Se, Rb, Sr, Cd, and Pb, an 8-point calibration curve (1, 5, 10, 25, 50, 100, 250, and 500 µg/L) was generated with standard solutions prepared by properly diluting a multi-element stock solution (Periodic table mix 1, 10 mg/L, TraceCERT^®^, Sigma-Aldrich) in 2% HNO_3_. For Mo, an 8-point calibration curve (0.05, 0.25, 0.5, 1.25, 2.5, 5.0, 12.5, and 25 µg/L) was generated with standard solutions prepared by properly diluting a multi-element stock solution (Periodic table mix 2, 10 mg/L, TraceCERT^®^, Sigma-Aldrich) in 2% HNO_3_. For Fe, an 8-point calibration curve (50, 100, 250, 500, 1000, 1500, 2000, and 5000 µg/L) was generated by properly diluting a multi-element stock solution (Periodic table mix 1) and a single-element stock solution (Iron standard for AAS, 1000 mg/L, Fluka) in 2% HNO_3_. For Hg, a 4-point calibration curve (1, 2, 5, and 10 µg/L) was generated with standard solutions prepared by properly diluting a Hg stock solution (Mercury standard for ICP, 1000 mg/L, TraceCERT^®^, Sigma-Aldrich) in 5% HNO_3_ in borosilicate glass volumetric flasks. These calibration solutions were then diluted with the diluent solution as the samples. The elemental isotopes ^7^Li, ^27^Al, ^52^Cr, ^55^Mn, ^59^Co, ^65^Cu, ^66^Zn, ^75^As, ^82^Se, ^85^Rb, ^88^Sr, ^98^Mo, ^111^Cd, ^202^Hg, ^206^Pb, ^207^Pb, and ^208^Pb were measured for analytical determination, and the elemental isotopes ^45^Sc, ^89^Y, ^141^Pr, and ^175^Lu were monitored as IS.

After thorough homogenization on a vortex mixer, the diluted samples and calibration standards were presented to the ICP-MS instrument using a CETAC ASX-520 autosampler (Teledyne CETAC Technologies, Omaha, NE, USA).

For analytical quality control, two certified reference materials (CRM) were used: Cabbage Powder (BCR-679), and Lichen (*Pseudevernia furfuracea*) Powder (BCR-482), both from the European Commission’s Joint Research Centre (JRC). The CRM were analyzed under the same pre-treatment and analytical procedure as the samples.

## 3. Results

The results obtained for the CRM (analytical quality control) are presented in Appendix A. CRM analysis showed an accuracy of 92–120% for all the elements with certified value in BCR-679, and 75–95% in BCR-482 (except Mo). The limits of detection (LD) and the limits of quantification (LQ) are presented in Appendix A.

Table 1 summarizes the results obtained for the essential trace elements Cr, Mn, Fe, Co, Cu, Zn, Se, and Mo in the analyzed carrageenan and gum samples. Table 2 summarizes the results obtained for the non-essential and toxic elements Li, Al, As, Rb, Sr, Cd, Hg, and Pb. Results are presented as the mean (standard deviation, SD) of three independent determinations and are expressed in µg/g. 

### 3.1. Essential Trace Elements Content

Chromium—the mean (SD) content found in carrageenans varied between 0.187(0.010) and 2.122(0.064) μg/g, with refined samples showing lower levels. Gums 11 [Xanthan] and 13 [Gellan] showed higher contents than the others, suggesting that the fermentation of saccharides by the stems leads to higher levels of Cr.

Manganese—the 5 [SRCι] sample had the highest content with 23.16(0.31) μg/g, but no trend was found between carrageenan processing and Mn level. In general, gums had lower contents than the carrageenans.

Iron—the content in the carrageenan samples varied between 1.35(0.88) and 189(11) μg/g, with SRC showing higher levels, with the exception of sample 6 [RCκ]. Regarding the gum samples, gum 11 [Xanthan] had an Fe content of 123.5(8.9) μg/g, which is about 5 times higher than in gum 13 [Gellan], 10 times higher than in gums 9 [Guar] and 10 [Acacia], and 20 times higher than in gum 12 [Tara], suggesting that fermentation of saccharides by the stems leads to higher levels of Fe in the gums.

Cobalt—carrageenans had higher Co contents than the gums. Sample 5 [SRCι] had a content of 1.80(0.05) μg/g, significantly higher than the others.

Copper—the content was generally less than 1.0 μg/g, with the exception of samples 2 [SRCκ], 5 [SRCι], 9 [Guar], 10 [Acacia], and 12 [Tara], the latter showing the highest Cu content: 3.18(0.19) μg/g.

Zinc—the content proved to be quite variable, ranging from 0.329(0.043) for sample 10 [Acacia] to 10.41(0.21) μg/g for sample 5 [SRCι].

Selenium—all analyzed samples showed a Se content below 1 μg/g, ranging from 0.077(0.027) in sample 10 [Acacia] to 0.63(0.10) μg/g in sample 7 [RCκ].

Molybdenum—contents were within the range of <LD (0.006) μg/g to 2.183(0.076) μg/g, for carrageenans 7 [RCκ] and 9 [Guar], respectively.

### 3.2. Toxic Trace Elements Content

Arsenic—the content in the gums was below the LD (0.047 μg/g) but was detectable in carrageenans. The highest content was found in sample 5 [SRCι], with 2.042(0.034) μg/g.

Cadmium—the content in the gums was below the LD (0.003 μg/g), with the exception of sample 9 [Guar], but detectable in all the carrageenans. In these samples, the levels varied between 0.022(0.002) μg/g in sample 1 [SRC mixture κ/λ] and 1.090(0.020) μg/g in sample 5 [SRCι].

Mercury—all samples had Hg levels below the LD (0.009 μg/g), with the exception of sample 11 [Xanthan] with 0.087(0.006) μg/g.

Lead—all samples showed detectable Pb levels, reaching the highest values in sample 5 [SRCι] with 0.606(0.095) μg/g, sample 3 [SRCκ] with 0.361(0.035) μg/g, and sample 12 [Tara] with 0.367(0.022) μg/g.

### 3.3. Other Trace Elements Content

Lithium—levels in gums were much lower than in carrageenans. The highest content, 23.6(1.4) μg/g, was found in sample 2 [SRCκ], while the remaining samples were generally below 1 μg/g. The results also indicated a Li content 10 to 20 times lower in refined samples when compared to unrefined ones.

Aluminum—the content was below the LD (5.1 μg/g) in two gums (samples 10 [Acacia] and 12 [Tara]), and below the LQ (15.5 μg/g) in samples 9 [Guar] and 13 [Gellan]. The carrageenans showed Al levels ranging from 31.6(1.2) μg/g in sample 7 [RCκ] to 188(19) μg/g in sample 2 [SRCκ].

Rubidium—content was significantly higher in carrageenans than in the gums, with sample 2 [SRCκ] having the highest content with 68.2(5.2) μg/g, and sample 11 [Xanthan] the lowest with 0.615(0.071) μg/g.

Strontium—gums globally showed lower contents. In the carrageenans, the levels ranged from 13.32(0.34) μg/g in sample 7 [SRCκ] to 136(10) μg/g in sample 2 [SRCκ].

## 4. Discussion

For carrageenans, the acceptable daily intake is 75 mg/kg bw, which means that a daily dose of 3.75 g is safe for a 50 kg person. However, according to EFSA panels [6], the estimated exposure to carrageenans is 10 g/day; i.e., 200 mg/kg bw per day for a 50 kg adult. Acceptable daily doses of gums are much higher, ranging from 128 to 430 mg/kg bw per day, which means that an intake of 6.5 to 21.5 g per day is acceptable for a 50 kg adult [1,2,3,4,5].

### 4.1. Essential Trace Elements

Table 3 presents the contribution of each sample to the U.S. Institute of Medicine (IOM) dietary reference intakes (DRI) [9,10]. For this calculation, a 50 kg adult male consuming the acceptable daily doses established by the EFSA for each food additive was used: 128–429 mg/kg per day for guar gum (279 mg/kg per day, the midpoint of the specified interval, was used for the calculations); 214 mg/kg per day for xanthan and gellan gums; 430 mg/kg per day for acacia gum; and 75 mg/kg per day for carrageenans [1,2,3,4,5,6].

#### 4.1.1. Chromium

Depending on its oxidation state, Cr can be an essential or highly toxic trace element. Trivalent Cr is an essential micronutrient as it is involved in glucose and insulin metabolism. It is also necessary for sucrose and fat metabolism [11]. In the hexavalent form, it is highly toxic, being a carcinogenic metal, as it can interfere with DNA synthesis and repair [12]. It may also have negative effects on several other macromolecules, causing genomic instability and reactive oxygen species (ROS) generation, leading to oxidative stress [13]. Thus, overexposure to Cr(VI) and its bioaccumulation can lead to kidney dysfunction, gastrointestinal disorders, dermatological diseases and increased incidence of certain types of cancer, including in the lungs, larynx, gallbladder, kidneys, testes, bones, and thyroid [14].

Daily exposure to Cr from dietary sources, excluding food supplements, is estimated to be around 0.1 mg/kg bw per day [15]. Adequate intake of Cr is set at 35 and 25 μg/day for adult men and women, respectively [10]. The UK Expert Group on Vitamins and Minerals estimates that an intake of around 0.15 mg/kg bw per day is expected to be safe [16].

The Cr content of the analyzed samples varied between 2.88(0.28) μg/g in sample 11 [Xanthan gum] and 0.102(0.036) μg/g in sample 10 [Acacia gum], which corresponds, respectively, to the intake of 88% and 6% (Table 3) of the daily requirement of a 50 kg adult male when consuming the acceptable daily dose of food additives. Unfortunately, the quantification technique used does not allow distinguishing between trivalent and hexavalent Cr. 

#### 4.1.2. Manganese

This trace element is essential for numerous vital processes, such as brain and neuronal development, and cognitive functioning [17]. There are several enzymes that require Mn as a cofactor, such as kinases, transferases, and decarboxylases. There are also some enzymes strictly dependent on Mn, such as glycosyltransferases. On the other hand, the accumulation of excessive levels of Mn in specific brain regions leads to neurotoxic effects. Thus, Mn intoxication results in neurological symptoms such as permanent tremors, motor incoordination, and memory problems [17].

Adequate Mn intake is set at 2.3 mg/day for adult male and 1.8 mg/day for adult female, and the Tolerable Upper Intake Level (UL) is 11 mg/day for adults 19 years of age and older [10]. The Mn content of the analyzed samples varied between 0.300(0.064) μg/g and 23.16(0.31) μg/g. The studied gums and carrageenans do not seem to contribute significantly to the daily intake of this element (Table 3).

#### 4.1.3. Iron

Iron is an essential micronutrient for humans as it is vital for cellular homeostasis, energy-producing processes (such as ATP), oxygen transport, and antioxidant activity [18]. Excessive levels of Fe in the body can have negative effects, and excessive intake interferes with the absorption of divalent trace elements [18]. It is known that Fe toxicity is mediated by the generation of ROS, which induces inflammation, promoting tissue damage and organ dysfunction [19,20,21]. Individuals ingesting 20–60 mg/kg of elemental Fe may develop nausea and vomiting but are at low risk of toxicity [22].

The [SRCκ] samples 1–3 stood out for their higher Fe content, with a maximum of 189(11) μg/g in sample 1. This content is about 3 times higher than that reported for certain non-heme Fe foods such as baked beans/lentils (0.05 mg/g) or white bread (0.04 mg/g) [23]. These carrageenan samples can contribute up to 9% of the daily Fe requirements of a 50 kg adult male (Table 4).

#### 4.1.4. Cobalt

This trace element is essential for human health in the form of cobalamin (vitamin B12), necessary for erythropoiesis, amino acid and nucleic acid metabolism, methionine synthesis, and conversion of methylmalonyl-CoA to succinyl-CoA in the Krebs cycle [24]. Excessive intake of Co can induce toxicity [25], with damage to the pancreas (in particular the cells that produce insulin and glucagon), respiratory tract, gastrointestinal tract, thyroid, and skin [26,27].

According to a survey carried out in France, the average exposure of the population to Co was estimated at 0.18 μg/kg bw per day in adults and 0.31 μg/kg bw per day in children [28], a value well below the lower limit of the tolerable daily intake set by the Agence Française de Sécurité Sanitaire des Aliments (1.6 μg/kg bw per day) [29].

Globally, carrageenans had higher Co contents than gums. Sample 5 [SRCι] had a Co content of 1.795(0.046) µg/g, significantly higher than the others. Given that the intake of 3.75 g of carrageenans by a person weighing 50 kg is considered safe [6], this would correspond to a Co dose of 6.7 µg/day (0.13 μg/kg bw/day), which is almost as much as the average daily exposure of the French population mentioned above [28]. In the remaining samples, the Co content was in line with that described for other foods, such as chocolate (0.139 mg/kg), offal (0.091 mg/kg) and butter (0.046 mg/kg) [28].

#### 4.1.5. Copper

This trace element plays an important role in human metabolism. It is a cofactor of numerous enzymes (e.g., cytochrome C oxidase, superoxide dismutase, tyrosinase), so it is essential for many physiological processes such as iron homeostasis, angiogenesis, neurotransmitter biosynthesis, immune function, and energy metabolism [30]. With excessive exposure, Cu can cause harm on multiple levels, including anemia, liver and kidney damage, gastrointestinal irritation, and cardiovascular or CNS disorders [30]. Chronic overexposure can also contribute to the development of Alzheimer’s disease.

The recommended daily intake of Cu is 900 μg/day for both adult men and women, while the UL for adults (over 19 years) is 10 mg/day [10]. The maximum Cu content found in the studied samples was 3.18(0.19) μg/g—in sample 12 [Tara gum]—so, the consumption of these food additives represents a negligible contribution to the daily intake of Cu.

#### 4.1.6. Zinc

This micronutrient performs essential biological activities at three levels: structural, enzymatic, and regulatory. Zinc is necessary for the proper functioning of the immune system; it is a key component of several enzymes (with a catalytic or structural role) that participate in the metabolism of proteins, carbohydrates, lipids, and nucleic acids; it is indispensable to the antioxidant and anti-inflammatory physiological function [31]; and it is involved in biological processes such as growth and development of neurological function. It also plays a key role in cell cycle regulation, DNA repair and replication, cell proliferation and differentiation, and apoptosis [32].

The Recommended Dietary Allowance (RDA) for Zn is 11 mg for adult male and 8 mg for adult female, increasing during pregnancy and lactation to 11 mg and 12 mg, respectively. The UL for Zn is 40 mg/day for adults over 19 years of age [10]. The highest Zn content found in the analyzed samples was 10.41(0.21) μg/g—in sample 5 [SRCι]—a negligible amount when compared to the recommended daily intake.

#### 4.1.7. Selenium

Selenium is an important micronutrient, necessary for the synthesis and proper functioning of several proteins [33]. After absorption, it is incorporated into several selenoproteins that are involved in important metabolic pathways, such as the synthesis of thyroid hormones and DNA. In particular, Se is involved in antioxidant defense as a key component of the antioxidant enzymes glutathione peroxidase (GPX) and thioredoxin reductase [33]. Deficiency impairs fertility and reproduction and increases the risk of cancer and viral infections [34]. Selenium is also known to promote the decreased heavy metal toxicity through the formation of selenides and metal-protein bonds [34].

Sources of Se for humans are dairy products, cereals, rice, and tuna. The fortification of food with Se must be done in a judicious and regulated manner, as it has a relatively narrow safety margin, i.e., the difference between the required daily dose and the toxic dose is small compared to the other essential trace elements [33]. The RDA of Se is 55 μg/day for individuals aged 14 years and older, with a UL of 400 μg/day [9]. That is, the tolerable upper intake limit is only about 7 times higher than the recommended daily intake.

The highest Se content found in the studied samples was 0.63(0.10) μg/g in sample 7 [RCκ]. The contribution of these additives to the daily requirements of Se seems to be very low (Table 3). The Se contents found are quite low when compared to Se-rich foods such as tuna (which has an average content of 6,8 μg/g) [35], but higher than those of polysaccharide-rich foods, such as rice, where the average Se content has been reported as 0.025(0.011) μg/g [36].

#### 4.1.8. Molybdenum

This trace element is present in the active center of some flavin-containing enzymes. It is also a cofactor of some other enzymes [37]. Excess Mo can increase the xanthine oxidase activity, leading to an overproduction of uric acid, which can cause gout and chronic kidney disease [38]. The RDA for both adult men and women is 45 μg/day, while the UL for Mo in adults over 19 years of age is 2 mg/day [10]. Of the analyzed samples, 9 [Guar gum] had a Mo content much higher the others with 2.183(0.076) μg/g, providing 68% of the daily Mo requirements for a 50 kg adult male (Table 3). The remaining gums and carrageenans had negligible amounts of Mo compared to sample 9 [Guar gum].

### 4.2. Toxic Trace Elements

The limits set by EU regulations for As, Pb, Cd, and Hg content in food additives depend on the type of additive. For gums, they are 3, 2, 1, and 1 μg/g, respectively, while for carrageenans they are 3, 5, 2, and 1 μg/g, respectively [39]. However, the EFSA Panel on Food Additives and Nutrient Sources added to Food (now the Panel on Food Additives and Flavourings) recognizes that “contamination at such levels could have a significant impact on the exposure to these elements, for which the exposures already are close to the health-based guidance values or benchmark doses (lower confidence limits) established by EFSA” [1,2,3,4,5]. All samples of gums and carrageenans analyzed had As, Pb, Cd, and Hg contents below the maximum levels established by EU regulations [39]. It should be noted that expert panels from various authorities have been reviewing exposure limit values for some of the most toxic elements such as As, Pb, Cd, and Hg [40,41,42,43].

Table 4 presents the contribution of each sample to the tolerable daily/weekly intake (TDI/TWI) set by the EFSA [40,41,42,43,44] or provisional chronic oral reference dose (p-RfD) set by the EPA [45,46]. Calculations were performed as described above for essential trace elements. As the EFSA does not have a TDI or TWI for As and Pb, the EFSA lower limit of the benchmark dose level for a 1% increased risk (BMDL_01_) of lung, skin, and bladder cancer of 0.3 μg/kg bw per day for inorganic As [40], and BMDL_10_ of 0.63 μg/kg bw per day associated with an increased risk of chronic kidney disease for Pb [41] were used.

#### 4.2.1. Arsenic

Arsenic is a metalloid that is highly harmful to human health. It is a carcinogen as it can interfere with DNA synthesis and repair [47]. Bioavailability and physiological/toxicological effects depend on the chemical form, with organic As(III) being the least toxic, and inorganic As(V) being the most toxic [47]. It may also have negative effects on several other macromolecules, leading to capillary endothelium damage, thiol binding (GSH conjugation), inhibition of ATP production, and alterations in neurotransmitter homeostasis [47]. Arsenic can lead to cardiovascular disorders, skin and hair changes, CNS disorders, and liver disease [47]. It is also associated with the dysfunction of numerous vital enzymes for humans. Thus, inorganic As, in addition to increasing the likelihood of skin, bladder, lung, kidney, and liver cancer, also increases the risk of developing diabetes, cardiovascular, neurological, gastrointestinal, kidney and liver diseases, in addition to having negative effects on reproduction, among others [48].

The highest As content was found in sample 5 [SRCι] with 2.042(0.034) μg/g, while all gum samples had As content <0.047 μg/g (LD). As shown in Table 4, some of the carrageenan samples can contribute significantly to the daily dietary exposure to As (5–51% of the BMDL_01_ established by the EFSA).

#### 4.2.2. Lead

Lead affects almost every organ in the body. With chronic exposure, it can cause CNS and hematological disorders (especially anemia), gastrointestinal cramps, liver and kidney damage, reproductive system disorders, decreased lung function, and cardiovascular dysfunction [49]. It interferes with many macromolecules, increasing the production of inflammatory cytokines (e.g., IL-1β, TNF-α, and IL-6) in the CNS and serum levels of endothelin-1 and erythropoietin. It inactivates δ-ALAD and ferrochelatase (thus inhibiting heme biosynthesis and causing hypochromic anemia), and reduces the activity of the most important antioxidant enzymes: superoxide dismutase (SOD), catalase (CAT), and glutathione peroxidase (GPX) [50]. It may therefore disturb the balance of the oxidant-antioxidant system and thus induce inflammatory responses in various organs [48].

In our study, detectable levels of Pb were found in gums and carrageenans, reaching the highest values in samples 12 [Tara gum] and 5 [SRCι] with 0.367(0.022) and 0.606(0.095) μg/g, respectively. The contribution of the analyzed samples to the BMDL_10_ established by the EFSA for Pb appears to be low, with a maximum of 7% for sample 5 [SRCι] (Table 4).

#### 4.2.3. Cadmium

Cadmium is highly toxic to plants and animals. It is a carcinogenic metal for humans, as it can interfere with DNA synthesis and repair, increasing the risk of developing cancer. Its toxic effects are manifested in various organs of the human body, and can lead to damage to bones, kidneys, liver, pancreas, thyroid, lung, and CNS [51]. It can also have negative effects on several other macromolecules, which can lead to cell apoptosis, endoplasmic reticulum stress, dysregulation of Ca, Zn, Cu, and Fe homeostasis, ROS generation, and impairment of phosphorylation cascades, among other harmful effects [48,52].

Sample 5 [SRCι] had the highest Cd content with 1.090(0.020) μg/g, while all but one gum sample (9 [Guar]) had a Cd concentration <0.003 μg/g (LD). As shown in Table 4, consumption of these samples at acceptable daily doses by a 50 kg adult male would represent an oral exposure to Cd always lower than 23% of the TWI established by EFSA. Despite being well within Cd limits established by EU regulations [39], the carrageenan samples analyzed showed higher levels than other foods such as crustaceans and mollusks (0.167 μg/g), cereal bars, and chocolate (respectively, 0.030 and 0.029 μg/g) [28].

#### 4.2.4. Mercury

Mercury is a potent neurotoxin. Organic species (notably the methyl-Hg found in fish) are significantly more toxic [53,54]. It particularly affects the CNS, but also the kidney, gastrointestinal tract (ulceration), and liver [55]. It can inhibit several enzymes, leading to the production of ROS and reducing aquaporin messenger RNA expression [56]. Symptoms triggered by chronic Hg overexposure include irritability, depression, anxiety, stress sensitivity and emotional lability, muscle weakness, mood swings, aggressiveness, and dementia [48,57].

The Hg content found in all analyzed samples was lower than the LD (0.009 μg/g), with the exception of sample 11 [Xanthan gum], with 0.087(0.006) μg/g. Unfortunately, the ICP-MS method used in this study only allowed the determination of the total Hg content in the samples and not the discrimination between organic and inorganic Hg.

### 4.3. Other Trace Elements

#### 4.3.1. Lithium

Lithium continues to play a major role in the treatment of some psychiatric disorders, in particular the management of manic episodes of bipolar disorder [58]. Recent studies indicate that Li may have neuroprotective effects [59], although the mechanism of action is not fully understood. Li also appears to play a positive role in human nutrition, as it has been shown to enhance the transport of vitamin B12 and folate to the brain and is believed to have anti-aging properties [60]. A provisional recommended daily intake of 14.3 μg/kg bw has been suggested for adults (715 μg/day for a 50 kg person) [61], while the EPA has a p-RfD of 2 μg/kg bw per day (100 μg/day for a 50 kg person) [46]. The studied gums and carrageenans do not seem to contribute significantly to the daily Li intake, except in the case of sample 2 [SRCκ], which may represent 89% of the p-RfD set by the EPA (Table 4).

#### 4.3.2. Aluminum

Currently, no beneficial effect of Al on human health is described in the literature. It is a possible toxic element and food is the main source of exposure. Several studies have shown that excess Al in the diet can lead to chronic intestinal problems such as irritable bowel syndrome, abdominal swelling, and poor digestion [62]. Overexposure to Al is particularly problematic when associated with chronic kidney disease [62]. Al body overload is associated with inhibition of P, F, and Ca absorption. In addition, it can interfere with several enzyme systems and damage cell membranes, having particularly harmful effects on certain tissues, namely bone and nervous tissue [63]. It can also cause hypochromic anemia [63].

Estimates of Al intake from whole diets have ranged from about 1 to more than 20 mg/day [64,65]. According to Arnich et al. [28], the average exposure of the French population is estimated at 40.3 μg/kg bw per day in adults and 62.2 μg/kg bw per day in children. Several of the analyzed samples showed Al contents between 100 and 200 μg/g. A 50 kg adult male consuming the acceptable daily doses of the analyzed carrageenan samples would be exposed to 2–10% of the EFSA-defined TWI (Table 4).

#### 4.3.3. Rubidium

Rubidium is considered a “nontoxic” element. It is found in fruits and vegetables at levels between 5 and >60 μg/g [66]. The Rb content in the carrageenans samples analyzed was significantly higher than in the gums, with sample 2 [SRCκ] having the highest level: 68.2(5.2) μg/g.

#### 4.3.4. Strontium

Strontium is also not considered an essential element, but some studies point to possible benefits in bone strength [67,68]. On the other hand, there are no known harmful effects of stable Sr on humans at levels typically found in the environment. A p-RfD has been estimated at 0.6 mg/kg bw per day [45]. Globally, the gums had a lower Sr content than the carrageenans. The lowest content in the carrageenans was found in sample 7 [SRCκ] and the highest in sample 2 [SCRκ], with 13.32(0.34) and 136(10) μg/g, respectively. As shown in Table 4, the analyzed samples do not seem to contribute significantly to daily intake of Sr.

## 5. Conclusions

Carrageenans and gums are food additives commonly used in a wide variety of food products. The present study is only a punctual survey, giving an instant picture of trace elements levels in particular batches of products on the market. It should be followed by broader periodical studies in the future.

Although a relatively small number of samples were analyzed, this study showed that the trace element content could vary widely between carrageenans and gums.

The results obtained also demonstrate that the daily consumption of these food additives at the acceptable levels established by the EFSA can significantly contribute to the daily requirements of some trace elements, particularly Cr and Mo.

Additionally, all analyzed samples showed As, Cd, Pb, and Hg contents well within the limits established by current EU regulations. However, if the estimated exposure to carrageenans reaches the highest value of 200 mg/kg bw per day, it can be a significant source of As and Cd.

More comprehensive studies are needed to validate the results obtained in this study and to evaluate the real contribution of these food additives to the daily intake of essential, non-essential, and toxic trace elements.

## Figures and Tables

**Table 1 foods-12-01408-t001:** Mean (SD) content (µg/g) of essential and probably essential trace element in carrageenan and gum samples.

Sample	Cr	Mn	Fe	Co	Cu	Zn	Se	Mo
**Carrageenans**								
1 SRC mixture κ/λ	0.710 (0.070)	3.31 (0.15)	189 (11)	0.327 (0.072)	0.45 (0.13)	3.03 (0.27)	0.093 (0.028)	0.072 (0.003)
2 SRCκ	1.15 (0.13)	11.04 (0.83)	184 (16)	0.649 (0.066)	1.24 (0.29)	2.30 (0.21)	0.247 (0.029)	0.155 (0.016)
3 SRCκ	2.122 (0.064)	7.535 (0.033)	152 (17)	0.845 (0.012)	0.94 (0.10)	6.02 (0.14)	0.474 (0.015)	0.045 (0.013)
4 SRCι	1.294 (0.077)	4.540 (0.039)	49.77 (0.49)	0.149 (0.004)	0.354 (0.079)	0.757 (0.083)	0.405 (0.035)	0.068 (0.022)
5 SRCι	0.547 (0.129)	23.16 (0.31)	127.2 (6.7)	1.795 (0.046)	1.37 (0.11)	10.41 (0.21)	0.360 (0.028)	0.039 (0.002)
6 RCκ	0.384 (0.045)	13.30 (0.36)	156.1 (5.9)	0.739 (0.068)	0.315 (0.006)	5.47 (0.16)	0.453 (0.064)	0.033 (0.005)
7 RCκ	0.187 (0.010)	0.300 (0.064)	1.35 (0.88)	0.082 (0.014)	0.234 (0.022)	0.446 (0.033)	0.63 (0.10)	<LQ
8 RCι	0.455 (0.078)	7.58 (0.21)	63.4 (5.1)	0.514 (0.055)	0.425 (0.006)	2.24 (0.12)	0.504 (0.035)	0.031 (0.001)
**Gums**								
9 Guar	0.187 (0.047)	3.73 (0.11)	13.6 (1.9)	0.068 (0.002)	1.758 (0.081)	6.061 (0.056)	0.245 (0.005)	2.183 (0.076)
10 Acacia (Arabic)	0.102 (0.036)	4.96 (0.24)	16.1 (1.7)	0.086 (0.003)	1.253 (0.056)	0.329 (0.043)	0.077 (0.027)	<LD
11 Xanthan	2.88 (0.28)	3.58 (0.22)	123.5 (8.9)	0.204 (0.012)	0.85 (0.19)	2.86 (0.29)	0.050 (0.031)	0.085 (0.009)
12 Tara	0.188 (0.043)	0.78 (0.02)	5.32 (0.67)	0.014 (0.004)	3.18 (0.19)	4.10 (0.45)	0.170 (0.023)	0.086 (0.001)
13 Gellan	2.55 (0.28)	2.43 (0.16)	24.1 (1.4)	0.040 (0.003)	0.220 (0.054)	0.599 (0.086)	0.082 (0.030)	0.066 (0.005)

LD (limit of detection; µg/g): Cr = 0.025; Mn = 0.029; Fe = 0.23; Co = 0.002; Cu = 0.033; Zn = 0.034; Se = 0.022; Mo = 0.006. LQ (limit of quantification; µg/g): Cr = 0.074; Mn = 0.086; Fe = 0.69; Co = 0.005; Cu = 0.099; Zn = 0.10; Se = 0.067; Mo = 0.018.

**Table 2 foods-12-01408-t002:** Mean (SD) content (µg/g) of non-essential and toxic trace element in carrageenan and gum samples.

Sample	Li	Al	As	Rb	Sr	Cd	Hg	Pb
**Carrageenans**								
1 SRC mixture κ/λ	0.090 (0.006)	79.1 (3.1)	< LD	18.68 (0.39)	80.0 (5.7)	0.022 (0.002)	<LD	0.064 (0.013)
2 SRCκ	23.6 (1.4)	188 (19)	0.492 (0.025)	68.2 (5.2)	136 (10)	0.168 (0.013)	<LD	0.066 (0.002)
3 SRCκ	0.282 (0.019)	142 (16)	1.689 (0.018)	16.41 (0.10)	104.0 (1.8)	0.604 (0.012)	<LD	0.361 (0.035)
4 SRCι	1.52 (0.19)	166 (14)	0.208 (0.021)	26.52 (0.48)	58.9 (1.1)	0.122 (0.004)	<LD	0.041 (0.001)
5 SRCι	0.240 (0.008)	138 (16)	2.042 (0.034)	11.43 (0.10)	110.4 (2.2)	1.090 (0.020)	<LD	0.606 (0.095)
6 RCκ	0.131 (0.019)	144 (22)	0.252 (0.019)	11.01 (0.24)	44.7 (1.1)	0.152 (0.003)	<LD	0.102 (0.004)
7 RCκ	0.103 (0.004)	31.6 (1.2)	0.216 (0.033)	14.64 (0.82)	13.32 (0.34)	0.012 (0.003)	<LD	0.052 (0.001)
8 RCι	0.137 (0.003)	66.4 (4.0)	0.225 (0.019)	8.65 (0.20)	83.2 (1.1)	0.233 (0.005)	<LD	0.121 (0.002)
**Gums**								
9 Guar	<LD	<LQ	<LD	4.26 (0.12)	9.91 (0.28)	0.018 (0.001)	<LD	0.028 (0.002)
10 Acacia (Arabic)	<LQ	<LD	<LD	9.90 (0.49)	67.0 (3.4)	<LD	<LD	<LQ
11 Xanthan	0.080 (0.005)	118.0 (6.6)	<LD	0.615 (0.071)	34.2 (2.6)	<LD	0.087 (0.006)	0.039 (0.013)
12 Tara	<LD	<LD	<LD	2.921 (0.046)	1.14 (0.10)	<LD	<LD	0.367 (0.022)
13 Gellan	0.068 (0.001)	<LQ	<LD	4.041 (0.052)	7.74 (0.24)	<LD	<LD	0.030 (0.002)

LD (limit of detection; µg/g): Li = 0.018; Al = 5.1; As = 0.047; Rb = 0.008; Sr = 0.081; Cd = 0.003; Hg = 0.009; Pb = 0.007. LQ (limit of quantification; µg/g): Li = 0.053; Al = 15.5; As = 0.14; Rb = 0.023; Sr = 0.24; Cd = 0.009; Hg = 0.028; Pb = 0.020.

**Table 3 foods-12-01408-t003:** Contribution (%) of different food additives to the Dietary Reference Intake (DRI) of essential trace elements.

Samples	Cr	Mn	Fe	Co *	Cu	Zn	Se	Mo
	**Carrageenans**							
1 SRC mixture κ/λ	8	1	9	-	0	0	1	1
2 SRCκ	12	2	9	-	1	0	2	1
3 SRCκ	23	1	7	-	0	0	3	0
4 SRCι	14	1	2	-	0	0	3	1
5 SRCι	6	4	6	-	1	0	2	0
6 RCκ	4	2	7	-	0	0	3	0
7 RCκ	2	0	0	-	0	0	4	nd
8 RCι	5	1	3	-	0	0	3	0
	**Gums**							
9 Guar	7	2	2	-	3	1	6	68
10 Acacia	6	5	4	-	3	0	3	4
11 Xanthan	88	2	17	-	1	0	1	2
12 Tara **	-	-	-	-	-	-	-	-
13 Gellan	78	1	3	-	0	0	2	2

DRI (mg/day)—Cr: 0.035, Mn: 2.3, Fe: 8, Cu: 0.9, Zn: 11, Se: 0.055, Mo: 0.045. * Co is essential only as a component of cobalamin and therefore does not have a DRI of its own; ** There is no established acceptable daily dose for Tara gum; nd—not determined, because content below LQ.

**Table 4 foods-12-01408-t004:** Contribution (%) of different food additives to the health-based guidance values (HBGV) of non-essential/toxic trace elements.

Samples	Li	Al	As	Rb *	Sr	Cd	Hg	Pb
	**Carrageenans**							
1 SRC mixture κ/λ	0	4	nd	-	1	1	nd	1
2 SRCκ	89	10	12	-	2	4	nd	1
3 SRCκ	1	7	42	-	1	13	nd	4
4 SRCι	6	9	5	-	1	3	nd	0
5 SRCι	1	7	51	-	1	23	nd	7
6 RCκ	0	8	6	-	1	3	nd	1
7 RCκ	0	2	5	-	0	0	nd	1
8 RCι	1	3	6	-	1	5	nd	1
	**Gums**							
9 Guar	nd	nd	nd	-	0	1	nd	1
10 Acacia	nd	nd	nd	-	5	nd	nd	nd
11 Xanthan	1	18	nd	-	1	nd	3	1
12 Tara **	-	-	nd	-	-	nd	nd	-
13 Gellan	1	nd	nd	-	0	nd	nd	1

p-RfD (mg/day)—Li: 0.1, Sr: 30. TWI (mg/week)—Al: 50, Cd: 0.125, Hg (inorganic): 0.2. BMDL_01_ (mg/day)—As: 0.015. BMDL_10_ (mg/day)—Pb: 0.032. * No HBGV is defined for Rb; ** There is no established acceptable daily dose for Tara gum; nd—not determined, because content below LQ.

## Data Availability

All data related to this study are presented in the article or Appendix A.

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
