# Peer review of "Determination by ICP-MS of Essential and Toxic Trace Elements in Gums and Carrageenans Used as Food Additives Commercially Available in the Portuguese Market"

_foods, 2023, doi:10.3390/foods12071408_

Round 1

Reviewer 1 Report

Gums and carrageenans are widely used food additives. Thus, the potential risk assessment is important in the case of their use. Thus, the authors tried to provide data on the levels of toxic trace elements and evaluate the level of daily intake and safety. Recent studies show small amounts of additives or specific food products can be significant sources of contaminants. This way such studies are important in the case of human health and food safety and allow for much deeper risk assessment.

The manuscript is well-prepared, written and organized. The Introduction might be shorter; however, I don't insist. It contains all the necessary information.

The authors chose 13 gums and carrageenans samples available in Portugal.

The data are presented very well, and the methodology is well-planned. Starting from sample preparation till ICP determination, the methodology looks good. All information is given in detail, the method seems repeatable. In my opinion quality control results (% values of accuracy) should be given. The authors mentioned they validate the method, but no details are given.

The discussion is very long but deeply treats all metals. I don't think table 3 is necessary. Also, the data from tables 4 and 5 should be given in the text. But, maybe the authors should enucleate from the text the information on % established adequate intake levels for adults for each measured metal and give them in a table - it can be added in tables 1 and 2 or in a separate table.   Conclusions and References are appropriate.   The authors have completed their aims. In my opinion, the English language and style are fine, however, I am not a native speaker. The same, I don't feel qualified to check, and I can not detect plagiarism.

Reviewer 2 Report

The present study mentions determining essential and toxic trace elements in gums and carrageenan using ICP-MS as food additives.

The title of the study presents this study as if it were a review article. Therefore, the title needs to be changed.

Only general information is given in the abstract part. No data on results are presented, which should also be revised.

In the Introduction section, general information about these additives is given, but previous studies are not mentioned. This is another drawback. If this work is the first, the value of it should be emphasized by the work done.

Researchers have claimed that excessive consumption of these food additives may be risky for human health. For this, they need to prove this with the necessary calculations like THQ (Target Hazard Quotient), HI (Hazard Index) and cancer risk. It will improve the MS.

Round 2

Reviewer 1 Report

The manuscript was substantially corrected and improved due to the reviewers' remarks. The balance between both reviews was kept.

I'm still unable to detect plagiarism, and I must trust the publisher it was checked.